# Spatial Differences in the Effect of Communities’ Built Environment on Residents’ Health: A Case Study in Wuhan, China

**DOI:** 10.3390/ijerph19031392

**Published:** 2022-01-26

**Authors:** Man Yuan, Haolan Pan, Zhuoran Shan, Da Feng

**Affiliations:** 1School of Architecture and Urban Planning, Huazhong University of Science and Technology, Wuhan 430074, China; yuanman_aup@hust.edu.cn (M.Y.); 595484172phl@gmail.com (H.P.); 2College of Pharmacy, Huazhong University of Science and Technology, Wuhan 430074, China

**Keywords:** healthy city, built environment, community, spatial differences

## Abstract

After 40 years of reform and opening-up policies, urbanization in China has significantly improved residents’ living standards; however, simultaneously, it has caused a series of health problems among Chinese citizens. Communities’ built environment is closely related to their residents’ health. However, few studies have examined the spatial differences in the health effects of community-built environments. Based on a 2013 health survey of residents in 20 communities in Wuhan, this study uses multilevel linear models to explore the effects of the built environment on residents’ health, analyzing the differences in its health-effect within different types of communities. The results showed that there were significant differences in the self-rated health status of residents in different communities, with those in high-end communities reporting a higher self-rated health status. The effect of the built environment on the health of residents in different communities was found to be inconsistent. For instance, the effect of the built environment on low-end community residents was very significant, but it was not obvious for residents in high-end communities. There are significant community-specific differences in the health- effect of the built environment: in high-end communities, residents’ health status was mainly restricted by travel accessibility, while in low-end communities, residents’ health status was mainly restricted by the accessibility of health facilities. Therefore, this paper proposes a built-environment optimization strategy for different types of communities to provide valuable insights for healthy community planning from a policy perspective.

## 1. Introduction

The rapid progress of urbanization and industrialization have significantly improved residents’ living standards; however, deteriorating urban environments threaten residents’ health [1,2]. Although motorized transportation has changed people’s way of living [3], it may have also facilitated the prevalence of chronic diseases, obesity, and being overweight among residents of urban environments. The Chinese government attaches great importance to the health of its citizens; its 14th Five-Year Plan and the Outline of the Long-term Goals for 2035 require comprehensive promotion of the development of a Healthy China strategy. The outline of Healthy China 2030 emphasizes addressing residents’ health problems at a community scale; that is, it advocates for the improvement of communities’ built environment, fostering healthy behaviors of grassroots residents, and actively promoting the development of healthy communities, thus alleviating residents’ health problems.

After the housing reform, the urban housing paradigm changed from unit distribution to market circulation. The original multi-class dormitories and government offices began to collapse, while the commercialization of housing led residents of similar socioeconomic status to gather in similar communities; subsequently, the environmental differences between communities became increasingly obvious, and the residential space gradually showed a trend of stratification [4,5].

Unlike European and American countries, communities are subdivisions of urban streets and administratively formed towns in mainland China, and local governments have drawn geographical boundaries for communities. Residents’ committees are set up in communities as the grassroots management organizations in China. In Wuhan, for example, there are 13 municipal districts, 156 street offices, and 1431 community committees. Community is a spatial unit of residents’ daily lives, and the differences between its built environment and social environment may affect residents’ health status. However, few studies have explored the issue of community differences in the health-effect of the built environment, and it is difficult to provide adequate directives for the planning and construction of healthy communities.

Therefore, based on the health survey data of community residents in Wuhan, this study uses multilevel linear models to discuss the following. (1) Are there any differences in the health status of residents in different types of communities? (2) How do the built environment’s elements affect residents’ health status? (3) Are there differences in the way the built environment affects health in different types of communities?

## 2. Literature Review

### 2.1. Health Effects of the Built Environment

Scholars from various countries have conducted a series of studies on the correlation betweem the built environment and health [6,7,8]. For example, Cervero et al. [9,10] first proposed a 3D model of a built environment affecting road traffic and believed that density, diversity, and design would affect the number of trips, trip mode, and route choice. Subsequent studies supplemented destination accessibility and distance to transit, which resulted in a 5D model. Ewing et al. [10] divided the built environment into five dimensions: density and intensity, land use mix, road connectivity, street scale, esthetic quality, and functional structure. Lu and Tan [11] divided the built environment into space elements and placed elements to discuss its influence on physical activity. Wang et al. [12] differentiated the four planning elements of land use, spatial form, road traffic, green space, and open space and analyzed their effects on residents’ health. These studies indicate that communities’ built environment has a significant effect on residents’ health status, and this effect is multidimensional and complex.

In terms of community density, studies conducted in developed Western countries posit that a higher living density may promote residents’ physical activity, thus improving their health status [11]. Medical facilities are extremely important factors affecting health status (Frank et al., 2005). When residents are in an environment with a reasonable allocation of medical resources, their health status will be significantly improved [13]. When the accessibility and quality of urban parks, squares, green spaces, and fitness and leisure facilities increase, more residents can be attracted to carry out leisure physical activities, which is conducive to improving their health status [14,15]. Improved accessibility between the residential area and the traffic station can reduce road traffic and increase residents’ physical activity, which could improve their health level [16]. Road network intersection density represents road network connectivity; high connectivity can reduce the distance from origin to destination and provide a variety of road travel options, thus improving residents’ willingness to travel on foot and reducing health risks [17]. In the food environment, fast food stores, convenience stores, bakeries, and candy and nut stores are considered unhealthy food stores [18], while wet markets and large supermarkets are considered health food stores because they can provide a wide variety of fresh food options and improve residents’ dietary habits [19,20].

### 2.2. Health Impacts of Community Differentiation

The commercialization of housing has contributed to the stratification of living spaces, resulting in significant environmental differences in communities’ demographic structure, economic level, and health facilities [21]. For example, the old unit communities and institutional compounds show an aging trend; urban low-income groups have shifted to urban fringe communities, while the cost of enjoying open spaces, transportation facilities, and health facilities and services has increased, resulting in an imbalance in the availability of health resources due to urban residential differentiation [22]. Residents living in high-end communities have a more spacious and convenient physical environment (including parks, open spaces, and exercise areas), more adequate transportation facilities, and a better food environment, all of which are positive factors that promote residents’ health [23,24]. In addition, community differences can influence residents’ behavioral habits and lifestyles to some extent, thus bringing various environmental and health inequities [25].

Western scholars have studied different types of communities and found that there are community differences in the effects of the economic level, demographic structure, and community environment on residents’ health [26]. Fewer studies in China have explored differences in the effects of the built environment on the health status of residents in different types of communities. China is in a period of economic and social transition, and the social stratification and spatial differentiation of residents are becoming increasingly evident, with urban communities showing significant spatial differentiation in both social composition and built environment [27,28,29]. For example, urban public service facilities have shifted from a traditional balanced distribution to higher-income settlements [30]. Studying community differences in the health-effect of residence in Chinese society not only contributes to the literature on healthy cities but also has important implications for enhancing the community habitat and improving residents’ health.

## 3. Data and Methodology

### 3.1. Study Area and Data

As an important city in central China, Wuhan has experienced rapid economic and social development; its urban space has been expanding, and its residential space has shown a distinctly heterogeneous pattern [31]. The Fifth National Health Services Survey is a household survey conducted in Wuhan City in September 2013 using a multi-stage stratified whole-group random sampling method. To address ethical issues, this survey was approved by the then National Health and Family Planning Commission and the National Bureau of Statistics, with an approval date of 30 June 2013, and a project start and end date of July 2013 to December 2013, with an approval number of National Health and Family Planning Commission Ethics Review (2013) 65, the investigator went into the field, explained the background, nature and significance of the study, emphasized that it was for scientific research only, and obtained informed consent from the guardians of the participants to formally conduct the survey. The survey mainly collected data on respondents’ socio-demographic characteristics, including gender, age, household registration, marital status, employment, education, health insurance, and income.

The European five-dimension quality of life scale (EQ-5D) was used to measure health-related quality of life, including health-related behaviors, such as smoking, alcohol consumption, and physical activity. The Visual Analogue Scale (VAS) is a subscale of the EQ-5D used to measure the self-rated health status of the general population. The VAS shows respondents’ comprehensive evaluation of their self-rated health. A score of 100 represents the best health condition, and 0 represents the worst health condition.

In this study, we used secondary data from the Fifth National Health Survey in Wuhan, conducted in September 2013, on 1922 residents in 20 communities (Figure 1) that were selected as the study sample; 1764 valid responses were collected by excluding 158 respondents under the age of 18 and those with missing items in any of the socio-demographic characteristics or in the EQ-5D. Among the excluded respondents, 82 were female and 76 were male, and 72 were residents of high-end communities while 86 were residents of low-end communities. The average sample size (±standard deviation) per community is 88.2 ± 10.2, with the largest being 105 and the smallest being 70, a relatively even sample size.

### 3.2. Indicator Selection and Research Framework

The dependent variable in this study’s multilevel linear model is the residents’ self-rated health status level (continuous variable). The VAS score and utility value measure are important tools for describing residents’ quality of life, and their reliability and validity have been confirmed in prior studies [32]. Although respondents’ self-rated health is subjective, it can reflect residents’ true physical and psychological health level [33], as it has a good predictive capacity regarding mortality, physical aging, and health service level [34,35,36]. In this study, we summarized the existing literature, considered the availability of data, and proposed a research framework consisting of individual attributes and the community environment (Figure 2). In the multilevel linear model, three types of indicators were selected for individual attributes: first, physical attributes (gender and age). Age and gender are considered to be associated with health status [37]; second, socioeconomic attributes (education, employment status, per capita annual income, and per capita housing area). Education and employment can affect the health of the residents in various ways [38,39,40]. Residents with higher per capita annual income and larger per capita housing areas tend to have better health; finally, medical attributes (medical checkup status and health insurance status), health insurance and health checkups are associated with the health status of the residents [41,42]. The three categories of the community environment include health facilities, transportation facilities, and community density (health/unhealthy food store ratio, density of medical facilities, and parks and squares area). The layout of health facilities can greatly affect the health status of residents [43,44]. Second, transportation facilities (density of transportation stations and density of road intersections). Transportation facilities can increase both the physical activity and exposure of residents [45]; Finally, community density (building density and floor area ratio). The impact of community density on residents’ health varies between the East and West [46,47].

Personal attributes were obtained from household survey data and community environment data from 2013, matching the time of the survey. The community environment was measured by taking the center of the community as a round point, and the appropriate distance of 800 m for residents’ daily activities as a radius [48], and the buffer zone range was delineated to measure the elemental indicators. Food stores (fast food stores, fried chicken stores, cake stores, dessert stores, milk tea stores, convenience stores, vegetable markets, fruit stores, and hypermarkets), medical facilities, transportation stations, and fitness facilities were obtained from the point of interest (POI) data, road intersection density from the Wuhan city road traffic map, park squares from the Wuhan city land use map, and building density and volume ratio from the Wuhan City Baidu map’s building information (Table 1).

### 3.3. Community Type Classification

This study uses residential characteristics as the basis for community evaluation and type classification to facilitate the exploration of community differentiation in the built environment, which affects residents’ health status. There are three main characteristics of residential types: architectural, neighborhood, and locational characteristics [49,50,51]. Based on previous studies [52,53], an urban community type classification index system (Table 2) was developed to measure the quantitative scores of each index. For positive indicators, the larger the indicator value, the higher the score. For negative indicators, the smaller the indicator value, the higher the score. Finally, the total score of each community is calculated, and the communities with the top 50% of the total score are defined as high-end communities, while those with the bottom 50% are defined as low-end communities.

Subsequently, the 20 analyzed communities were divided into two types: high-end and low-end. Among them, the 10 high-end communities included commercial houses with good location conditions and unit communities with good supporting facilities, whereas the 10 low-end communities included commercial houses, old unit communities, and old traditional neighborhood communities.

### 3.4. Research Methodology and Model Construction

The health status of residents is often influenced by both individual attributes and the community environment, thus forming a two-level nested structure of resident-community in the study data. However, existing empirical studies often use single-level regression models, such as multiple linear regression and structural equation models, which violate the assumption of mutual independence of variables in regression [54], ignore the differences between groups (community environment) [55], and cannot accurately describe the built environment’s on residents’ health status [56]. In contrast, multilevel linear models are widely used because they can separate both levels (residents and communities) and describe the contribution of each level to explain the dependent variable separately [54,55,57].

Therefore, this study uses a multilevel linear model to explore the spatial differences on the effects of the built environment on residents’ health using HLM 6.08 software. We first constructed three types of sample data: full sample, high-end community sample, and low-end community sample, and performed the following operations on each of the three types of samples. First, dummy variables were set for the categorical variables using SPSS26 software. Second, covariance detection was performed on the independent variables to exclude those that could not enter the model due to the presence of covariance. Third, the data were imported into HLM for Windows, version 6.08. Manufacturer: Scientific Software International, Inc. (Skokie, IL, USA), and the null model analysis was conducted to determine the extent of between-group (community) differences and to assess the applicability of the multilevel linear model. Finally, the full model was constructed to explore the effects of each variable on residents’ health status at the individual attribute level and the community environment level.

The full multilevel linear model equation is as follows:

Resident-level regression equation:(1)$$Y_{ij}=\beta_{0j}+\sum_{n=1}^{N}\beta_{nj}*X_{nij}+r_{0j}$$

Community-level regression equation:(2)$$\beta_{0j}=\gamma_{00}+\sum_{m=1}^{M}\gamma_{0m}*X_{mj}+u_{0j}$$
(3)$$Y_{ij}=\beta_{0j}+\sum_{n=1}^{N}\beta_{nj}*X_{nij}+r_{0j}$$

In the formula, *i* and *j* represent the resident and community levels, respectively; $Y_{ij}$ indicates residents’ health status; $X_{nij}$ indicates the variables at the individual attribute level (gender, age, income, etc.), *n* indicates the number of resident-level variables; $X_{mj}$ indicates the variables at the community environment level (population density, density of medical facilities, etc.), m indicates the number of community-level variables; $\beta_{0j}$ is a random variable; $\beta_{nj}$ is the regression coefficient of individual attribute variables; $\gamma_{0m}$ is the regression coefficient of community environment variables; and $r_{0j}$ and $u_{0j}$ are the random errors at the resident and community levels, respectively.

### 3.5. Descriptive Statistics of the Sample

Regarding residents’ self-rated health status (Table 3), the average overall value was 82.02, the average health status of the high-end communities was 83.33, and that of the low-end communities was 80.73. A non-parametric test was applied to test the difference between the health status of residents in the two types of communities, which showed a *p*-value of 0.000, while the significance level was set at 0.05. This indicates that there are significant differences in the health status of residents in different types of communities.

In terms of individual attributes, the percentage of women in the total sample was 54.2%. The average age of the residents surveyed was 47.7 years old, with most respondents (47.8%) being 40–60 years old. Compared with high-end communities, the proportion of elderly people (compared with that of young and middle-aged people) was significantly higher among residents of low-end communities. In the overall sample, 75% of respondents had a high school education or higher, while this value was 86% for high-end communities and 69.3% for low-end communities. Approximately 80% of respondents had medical insurance, but only 47.8% had undergone a medical checkup in the past year. The proportion of medical checkups was significantly lower in the low-end communities than in the high-end communities, which were 38% and 56.8%, respectively. Respondents’ average annual per capita household income was 39,900 CNY, with the highest percentage of respondents (39.3%) reporting an income between 10,000 and 30,000 CNY. The economic situation of residents in high-end communities was better, with 62% of them reporting an annual per capita household income above 30,000 CNY, while only 45.2% of respondents in the low-end communities reported such income. In terms of housing area per capita, respondents’ mean housing area per capita was 46.78 m^2^. Additionally, 36% of respondents in the overall sample and 39.8% in high-end communities reported a housing area per capita of 30–60 m^2^. Lastly, 38.7% of respondents in low-end communities reported a housing area per capita below 30 m^2^.

With regard to community attributes, all the indicators were continuous variables. In terms of the medical environment, the mean value of medical facility density in the buffer zone of the sample communities was 12.8 units/km^2^, while the density of medical facilities in high-end communities was significantly higher than that in low-end communities. The mean value of park square area was 8.3 ha, while that in high-end communities was 9.33 ha, and that in low-end communities was 7.28 ha. In terms of travel accessibility, the mean value of traffic station density in the sample communities was 6.01/km^2^, and the mean value of intersection density was 11.77/km^2^. Low-end communities have higher traffic station density and intersection density than high-end communities, which may be due to the fact that most of the low-end communities are historical communities that occupy a better location in the city due to their earlier construction [58]. In terms of community density, the mean value of building density in the sample communities was 0.286, and the mean value of the floor area ratio was 1.45. Low-end communities have higher building densities and lower plot ratios compared with high-end communities. The rest of the indicators were tested non-parametrically, and there was no significant difference between the two types of communities.

## 4. Results

### 4.1. Community Effects Test

This study first constructed null models of the community-built environment with residents’ health status for three types of samples. Only the dependent variables were substituted, and no independent variables were substituted to determine whether the samples were suitable for multilevel linear models and to analyze the extent to which differences in residents’ health status originated from community differences. All three models were tested using the chi-square test with a *p*-value of 0.000, indicating that the models passed the test and that the between-group variance (community) of each model was greater than its standard error, which indicates that the differences in the health status of the population originated largely from community differences [54]; however, specifically for each model, there were more significant differences. As shown in Table 4, for the total model, based on the between-group variance and within-group variance, the interclass correlation coefficient was calculated to be 0.077, which is greater than 0.059, showing a moderate correlation [59]. That is, the community level explains 7.7% of the variance in residents’ health status. For the low-end community model, the interclass correlation coefficient was 0.114. Conversely, the interclass correlation coefficient for the high-end community model was only 0.026. This indicates that the differences in the health status of residents in high-end communities originated from a small proportion of community differences and were mainly influenced by differences in individual attributes and other factors. In addition, although the interclass correlation coefficient of the high-end community model was less than 0.059, showing a low correlation [59]. The sample data itself is a hierarchically nested structure, so using a multilevel linear model is still more reliable than a single-layer linear model.

### 4.2. Influence of Individual Characteristics

The analysis in Table 4 shows that the elements of individual characteristics, with the exception of gender and annual per capita household income, have a significant effect on the health status of the overall sample. To some extent, the health status of the population decreases with age [37]. Education is positively associated with health status, which may be due to the fact that residents with higher education have a healthier lifestyle and possess more health care knowledge [38]. Employment status shows that employed residents have better health than retired residents, whereas unemployed residents have the worst health status. Studies have shown that retirement causes people to have significantly fewer social activities and leisure time, which impairs health [39], while unemployment makes residents have significantly lower income and generates great psychological stress, which has a negative effect on health [40]. The employed population has a stable income, a broader social environment, a healthier lifestyle, and work increases the population’s level of physical activity, all of which have a positive effect on health status [60]. Health insurance coverage is significantly associated with residents’ health status, and residents with health insurance enjoy a better health care environment, less medical burden, and greater health awareness, while their health status is relatively better [41]. Health checkups also have a positive effect on health status, as they allow residents to identify any ailments in a timely manner through health checkups [42]. Housing area per capita is negatively correlated with health status, and some studies have shown that the larger the housing area per capita, the more prone residents are to disease [61], but the exact reasons still need to be explored further. For residents of different types of communities, the influence of individual characteristics on health status shows some differences: residents of high-end communities are more significantly positively influenced by literacy than those of low-end communities; in low-end communities, residents with health insurance and health checkup habits tend to have better health status.

### 4.3. Influence of the Built Environment

Analysis of the full-sample regression results in Table 4 revealed that all community built-environment elements, except for volume ratio, had a significant effect on residents’ health status. Among them, the ratio of healthy/unhealthy food stores was significantly and positively correlated with health status, as the better the food environment around the community, the easier it is for residents to develop good eating habits and better health status [62]. The higher the density of medical facilities, the better the health status of residents, as these facilities can provide medical consultation, diagnosis, and treatment services for residents, significantly improving their health status [12]. In terms of travel accessibility, the higher the number of bus stops around the community, the better the health status of the residents, indicating that improving transit accessibility is beneficial for residents, as they travel via public transportation, which requires light physical activity [45]. An increase in road intersection density was linked to a decrease in self-rated health. A greater number of parks and public squares was associated with higher self-rated health status. Parks and public squares provide residents with good access to recreational activities and reduce their exposure to air, noise, and other pollution [63]. This study found that residents’ health will decline when building density and volume ratio increase.

### 4.4. Community Differentiation

Table 4 shows that after constructing the full model, the between-group variance (at the community level) of the total sample model and the low-end community model decreased by 9.784 and 21.982, respectively, whereas the between-group variance reduction ratios reached 76.8% and 92.8%, respectively. This illustrates that the community environment variables selected for this study explain the community-level differences in residents’ health status to a large extent [54]. For high-end communities, the interclass correlation coefficient (ICC) (2.649%) for the null model and the between-group variance reduction ratio (14.7%) for the full model were both small, suggesting that the differences in the health status of residents in high-end communities originated mainly from individual attribute differences and other factors, rather than at the community level.

For high-end communities, only three built-environment variables (traffic station density, intersection density, and floor area ratio) were significantly associated with self-rated health status. Combining this finding with the ICC and between-group variance reduction ratios described above, it is clear that the self-rated health status of residents in high-end communities is less affected by the community’s built environment. This is because the residents of the high-end communities analyzed in this paper were mainly young and middle-aged people, who generally have higher income and education, as well as stable jobs, enjoy a wider social circle, have better travel ability, and more areas for activities, while the small-scale community built environment does not have a significant effect on this group [46,64].

Furthermore, in high-end communities, the density of traffic stops and intersections both had a significant positive effect on health status, suggesting that the self-rated health status of residents in high-end communities is mainly governed by accessibility (Figure 3). On the one hand, according to the descriptive statistics of the sample, the density of road intersections is lower in high-end communities than in low-end communities, while the positive effect of intersection density on residents is mainly to promote physical activity [65] rather than to increase the negative effect of air pollution exposure [66]. On the other hand, residents of high-end communities have higher social status and better travel ability, and adequate transportation stations and well-developed road networks can facilitate residents’ travel to other areas for health-related resources to compensate for the lack of built environment in the community [46]. The negative effect of floor area ratio on the health status of residents in high-end communities was more significant than in low-end communities.

For low-end communities, all built environment variables, except for population density, had a significant effect on residents’ health status, and the regression coefficients were higher than those of high-end communities. Combining the ICC with the between-group variance reduction ratio, it is clear that the self-rated health status of residents in low-end communities is more influenced by the built environment around the community. Because of the limited radius of their daily activities, the most significant association with health status is the built environment around the community in a small area [46].

All health facility variables had a significant positive effect on residents of low-end communities. The health status of residents of low-end communities was mainly governed by the accessibility of health facilities (Figure 3). Unlike high-end communities, an increase in road intersection density will significantly reduce the health status of residents in low-end communities because most of them are located in the old city of Wuhan, and the density of the road network around the communities has reached a high level. Moreover, the intersection density reflects, to some extent, traffic volume, which in turn affects residents’ exposure to air pollution and health status [66]. Accordingly, when density increases, the negative effects will be much greater than the positive ones [67].

Building density has a significant negative effect on the residents of low-end communities, while the floor area ratio had no significant effect.

## 5. Discussion

### 5.1. High Density Built Environment

Among the health effects of the built environment, most of our findings are consistent with those of Western countries. However, some findings are of interest with regard to intersection density, building density, and floor area ratio. In contrast to prior Western studies [65], in this study, an increase in road intersection density was linked to a decrease in self-rated health, which was also observed in a study in Shanghai, China [66]. Wuhan is a megacity, similar to Shanghai, where the building and road densities are much higher than in western countries. The city has a considerable traffic flow every day, where motor vehicles stop and start at intersections in the idling stage. This will produce more pollution [68].

Empirical studies in developed countries reveal that a high-density residential environment helps promote physical activity and reduces health risks among residents [12,45]. However, this study found that residents’ health will decline when building density and volume ratio increase, consistent with a study of 480 community (village) committees in China [69] and a study in Wuhan [46,70]. Western countries are relatively sparsely populated, but according to a quantitative study of global urban density, the density of Asian cities is more than twice the world average [71], while the residential density of Wuhan’s main urban area is even higher. Taking the Hankou area where the COVID-19 outbreak occurred in 2020 as an example, its population density is 88,689 people/km^2^, the residential land density is 70.5%, and the building density is 67.5%. With a floor area ratio of 2.63, the building density is extremely high and there is a noticeable lack of roads and public space, which produces a series of problems, such as overcrowding, air pollution, and noise pollution.

### 5.2. Community Differentiation Problems

Over the 40 years of rapid urbanization in China, the pattern of spatial differentiation of living space has been clearly formed. New urban areas have a good built-up environment and bring together a number of high-end communities. According to the descriptive statistics of the sample, most of these communities are high-volume/low-density commercial housing, and the building density is much smaller than that of the low-end communities, so there is no disadvantage of high building density [67]. However, high plot ratios tend to cause a series of problems, such as reduced open space, a strong sense of depression, indifferent neighborhood relationships, and insufficient daylight and natural ventilation, which can damage residents’ physical and mental health [67,72,73].

The old city is also dotted with a lot of low-end communities. Based on the statistical description of the sample, most of these communities are low-volume/high-density and have a complex type of residence, a single type of land use, and a lack of activity space. High residential density not only affects the residents’ leisure-oriented physical activities and reduces their physical activity level but also hinders airflow and increases their exposure to air pollution [67].

By comparing the residents of the two communities, we found that most of the residents of high-end communities are young and middle-aged, with good travel ability, less influenced by the surrounding built environment, and can travel to other areas of the city to obtain health resources. Their health status is also mainly governed by “travel accessibility.” In contrast, most of the residents of low-end communities are middle-aged and old, with limited travel ability, and are greatly influenced by the surrounding built environment. Their health status is mainly governed by “health facilities’ accessibility.” However, there are often fewer public service facilities, such as medical facilities, green areas, and squares, in the old city, resulting in health inequity.

### 5.3. Policy Implications

Through the above findings, we observe that: the ultra-high density of China’s big cities, the incomplete facilities, inadequate functions, and insufficient public and social services in old neighborhoods are urgent problems to be solved. At present, China’s urbanization process has shifted from high-speed development to high-quality development and from incremental planning-oriented to stock planning-oriented. Based on this, the Chinese government has been carrying out large-scale urban renewal and has issued guidelines for the construction of 15-min living circles and complete residential communities (Figure 4). The government aims to promote the equalization of public services, eliminate health inequities, and enhance people’s happiness and sense of gain. After the outbreak of COVID-19, Wuhan quickly issued the “Wuhan Post-Epidemic Revitalization Plan (Three-Year Action Plan)”, which strives to optimize the distribution of medical facilities in the city and improve the primary health care and public health network to realize the “combination of normal time and epidemic.” In terms of urban space quality improvement, in combination with the previously introduced “Three-Year Action Plan for the Transformation of Wuhan’s Old Neighborhoods (2019–2022)”, Wuhan should optimize the land use structure of old urban areas by reducing the proportion of residential land and increasing the land for public service facilities, green areas and squares, road traffic, and others. It should also renew old cities based on low building density and avoid large-scale development.

### 5.4. Limitations

A limitation of this study is that it uses the place of residence as the geographic context of residents to study the built environment based on their self-rated health.

However, since the residents’ activity locations also include their workplaces, shopping places, and others, there is a deviation between the static residential units and the geographic context of their real experiences, creating an uncertain geographic context problem [74]. Therefore, the author will conduct research based on residents’ GPS trajectories or travel logs in future studies to obtain more accurate results.

## 6. Conclusions

In this study, we used multilevel linear models to explore the influence of the built environment on residents’ health status in 20 communities in Wuhan, and analyzed the differences in the effect in different types of communities. The main findings are as follows: (1) There are significant differences in the health status of residents in different types of communities, as residents in high-end communities have higher self-rated health levels. (2) The extent to which the differences in residents’ health status originate from the community varies; a large proportion of the differences in the health status of residents in low-end communities and a small proportion of those in the health status of residents in high-end communities originated from the community itself. (3) The degree of influence of the community-built environment on residents of different types of communities is inconsistent; specifically, residents of low-end communities belong to more disadvantaged groups and have a more limited travel radius, while their health behaviors and health status are more significantly influenced by the community-built environment. Further, residents of high-end communities have higher socioeconomic status and travel ability, and the effect of a small-scale community-built environment on them is not obvious. (4) There were differences in the action of the community-built environment on the residents’ self-rated health status. Residents of low-end communities are mainly constrained by the accessibility of health facilities and building density, while residents of high-end communities are mainly influenced by the accessibility of travel and floor area ratio.

Based on these findings, the conclusions of this study are as follows. A spatial differentiation pattern of living in Wuhan has clearly developed [27], and the community-built environment’s effects on the health status of residents may vary. Therefore, in the ongoing planning and construction of territorial spatial planning [75,76] and practical needs of residents of healthy communities for public service facilities in different types of communities should be taken into account. For example, most of the low-end communities that were analyzed in this study were old high-density neighborhoods where relatively more socially disadvantaged groups resided and whose health status is restricted by the accessibility of health facilities.

Within the context of urban renewal and the “Three Old” reform, we should strictly control building density, build pocket parks, administer the street space correctly, set up more community centers, improve the accessibility of health facilities, and provide residents with a high-quality food environment. Most of the high-end communities in this study comprised high-volume/low-density commercial housing, where residents have a relatively high socioeconomic status and their health conditions depend on travel accessibility. Therefore, following the perspective of urban incremental construction, land should be utilized in an appropriate and intensive manner [77], and the volume ratio should be strictly controlled to avoid excessive urban density. Open spaces should be laid out around the community, and urban bus stops should be installed to establish a well-connected transportation network that can promote residents’ interaction with each other [73] and ease their travel to other areas to access health facility resources [46].

## Figures and Tables

**Figure 1 ijerph-19-01392-f001:**
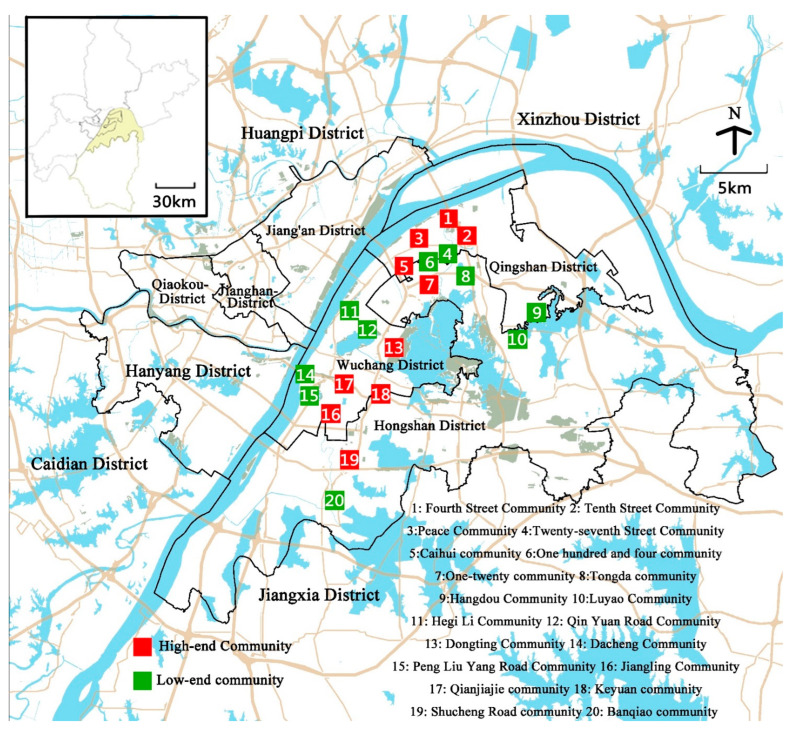
Distribution of case communities.

**Figure 2 ijerph-19-01392-f002:**
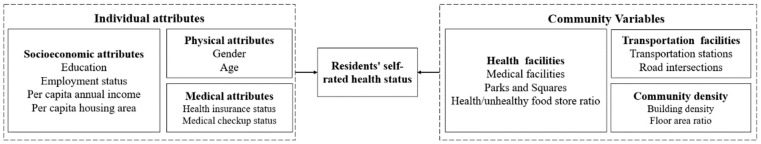
Research framework of the influence of personal attributes and community environment on residents’ self-rated health status.

**Figure 3 ijerph-19-01392-f003:**
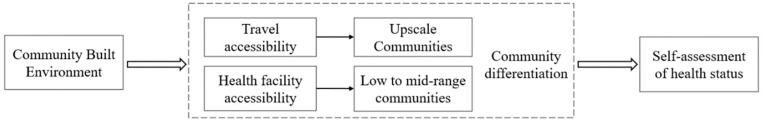
Community differences in the impact of a community-built environment on health status.

**Figure 4 ijerph-19-01392-f004:**
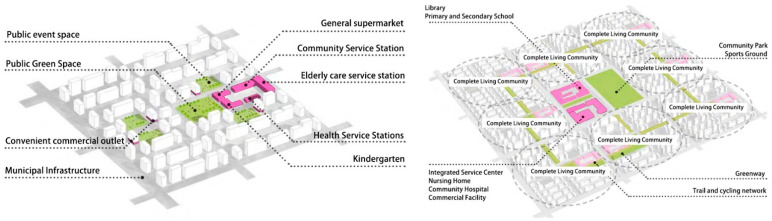
Complete residential community and 15-min living circle (Based on the *Complete Residential Community Guide*).

**Table 1 ijerph-19-01392-t001:** Source of Variables.

Type	Variable	Interpretation	Data Source
Health facilities	Health/unhealthy food store ratio	Ratio of the number of unhealthy food stores to healthy food stores in the buffer zone	poi
	Density of medical facilities	Density of medical service providers in the buffer zone	poi
	Parks and squares area	Parks and squares area in the buffer zone	Land Data
Transportation facilities	Density of traffic stations	Density of traffic stations in the buffer zone	poi
	Density of road intersections	Density of road intersections in the buffer zone	Road Data
Community density	Building density	Sample Community Building Density	Construction Data
	Floor area ratio	Sample Community Floor Area Ratio	Construction Data
Individual attributes	Gender	0 = male; 1 = female	survey
	Age	Respondents’ biological age	survey
	Education	0 = Junior high school and below; 1 = High School/Junior College; 2 = College/bachelor and above	survey
	Employment status	0 = Employed; 1 = retied, 2 = Unemployed	survey
	Health insurance	0 = No; 1 = Yes	survey
	Medical checkup	0 = No; 1 = Yes	survey
	Per capita annual income	Continuous Variables	survey
	Per capita housing area	Continuous Variables	survey
	Self-rated health status	Self-assessment value (0–100)	survey

**Table 2 ijerph-19-01392-t002:** Classification of Urban Communities.

Type of Features	Evaluation Indicators	Interpretation	Indicator Direction
Architectural Features	House price	Sale price per square meter of residential units, from Anjuke website data, with a score of 1–5 according to equal intervals	Positive
	Building Age	The time between the completion of the residence and the present, according to the equal interval, assigned 1–5 scores	Negative
Neighborhood Features	Green Environment	The green space rate within the 800 m buffer zone of the community is assigned 1–5 scores according to the equal interval	Positive
	Supporting facilities	The number of poi in the community’s 800 m buffer is assigned a score of 1–5 based on equal intervals	Positive
Location Features	Traffic Location	The Euclidean distance of the community from the nearest transportation station, according to the equal interval, assigning a score of 1–5	Negative
	Geographical location	Community distance from Wuhan central activity area in European style, according to the equal interval, assigned 1–5 scores	Negative

**Table 3 ijerph-19-01392-t003:** Statistical Description of the Individual Attributes of the Surveyed Residents and the Attributes of the Communities.

Variable	Definition and Units		Full Sample	Low-End Community Sample	High-End Community Sample
Gender	Male (%)		45.8	46.1	45.6
	Female (%)		54.2	53.9	54.4
Age	Age 18–25 (%)		6.2	6.3	6.2
	Age 25–40 (%)		30.3	26.7	33.5
	Age 40–60 (%)		47.8	50	45.7
	Over 60 years old (%)		15.7	17	14.6
Education	Junior high school and below (%)		25	30.7	14
	High School/Junior College (%)		36.5	40.1	33.2
	College/bachelor and above (%)		38.5	29.2	52.8
Employment status	Employed (%)		61.9	56.5	67
	Unemployed (%)		8.7	11.2	6.4
	Retired (%)		29.4	32.3	26.6
Health insurance	Yes (%)		79.3	76.8	81.5
	No (%)		20.7	23.2	18.5
Medical checkup	Yes (%)		47.8	38	56.8
	No (%)		52.2	62	43.2
Per capita annual income (10,000 CNY)	<1 (%)		6.7	8.8	4.8
	1–3 (%)		39.3	46	33.2
	3–5 (%)		26.9	24	29.5
	5–10 (%)		21.4	17.3	25.2
	>10 (%)		5.7	3.9	7.3
Per capita housing area (m^2^)	<30 (%)		35.5	38.7	32.6
	30–60 (%)		36	31.9	39.8
	>60 (%)		28.5	29.4	27.6
Density of medical facilities	Number per km^2^ in the buffer (units/km^2^)	Means	12.80	11.00	14.60
		Standard deviation	5.74	6.01	5.16
Health/unhealthy food store ratio	Ratio of healthy to unhealthy food stores in the buffer zone (%)	Means	37.97	38.2	37.77
		Standard deviation	13.82	15.80	12.49
Density of traffic stations	Number per km^2^ in the buffer (units/km^2^)	Means	6.01	7.51	4.51
		Standard deviation	2.87	2.06	2.85
Density of road intersections	Number per km^2^ in the buffer (units/km^2^)	Means	11.77	12.78	10.76
		Standard deviation	3.63	3.36	3.79
Parks and squares area	Area of the park square in the buffer zone (hm^2^)	Means	8.30	7.28	9.33
		Standard deviation	4.57	4.44	4.72
Building density	Building density in the community (%)	Means	28.64	30.8	26.50
		Standard deviation	10.17	10.96	9.45
Floor area ratio	Volume ratio in the community (dimensionless)	Means	1.45	1.26	1.64
		Standard deviation	0.53	0.47	0.51
Average self-assessed health status		Means	82.02	80.73	83.33
		Standard deviation	13.02	13.49	12.16
Sample size			1764	840	924

**Table 4 ijerph-19-01392-t004:** Regression Model Results.

Explanatory Variables			Full Sample	High-End Community Sample	Low-End Community Sample
Individual attributes	Gender (Refer to: male)	Female	0.608	0.992	0.037
	Age		−0.268 ***	−0.280 ***	−0.254 ***
	Education (Refer to: Junior high school and below)	School/Junior College	0.499	−0.088	0.771
		College/bachelor and above (%)	1.158 *	0.998 *	1.148
	Employment status (Refer to: Employed)	Retired	−2.867 ***	−3.150 ***	−3.408 **
		Unemployed	−6.716 ***	−3.819 *	−9.667 *
	Per capita annual income		0.101	0.131	0.122
	Per capita housing area		−0.047 ***	−0.037 ***	−0.056
	Medical checkup (Refer to: No)	Yes	0.835 ***	0.519 *	1.131 ***
	Health insurance (Refer to: No)	Yes	0.451 *	0.263	0.593 **
Environment Variables	Health/unhealthy food store ratio		0.812 ***	0.475	1.281 ***
	Density of medical facilities		1.606 ***	1.931	1.359 ***
	Parks and squares area		3.478 ***	2.587	3.909 ***
	Density of traffic stations		1.015 ***	1.848 **	0.359 ***
	Density of road intersections		−0.899 **	1.023 *	−1.291 **
	Building density		−0.331 ***	−0.256	−0.418 ***
	Floor area ratio		−0.903	−2.934 ***	−0.685 *
Null model	Variance between groups		12.727	3.39121	23.679
	Within-group variance		152.148	124.566	183.888
	ICC		7.713%	2.649%	11.408%
Complete model	Variance between groups		2.943	2.894	1.697
	Within-group variance		127.169	103.284	153.284
	ICC		2.264%	2.730%	1.094%
	Between-group variance reduction ratio		76.8%	14.7%	92.8%

Note: *, **, and *** are tests passed at the 0.1, 0.05, and 0.01 significance levels, respectively, interclass correlation coefficient (ICC) = between-group variance/(within-group variance + between-group variance); between-group variance reduction ratio = (null model between-group variance-complete model between-group variance)/null model between-group variance.

## Data Availability

The data presented in this study are available on request from the corresponding author.

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
