# Peer review of "Spatial Differences in the Effect of Communities’ Built Environment on Residents’ Health: A Case Study in Wuhan, China"

_ijerph, 2022, doi:10.3390/ijerph19031392_

Round 1

Reviewer 1 Report

Manuscript title: Spatial Differences in the Effect of Communities’ Built Environment on Residents’ Health: A Case Study in Wuhan, China

Manuscript number: 1504327

Journal: IJERPH

Authors: Man Yuan, Haolan Pan, Zhuoran Shan

This paper examines built environment attributes and self-rated health in 1,764 adults from Wuhan, China, and investigates associations in relation to two types of community that vary by amenity. It is good to see this type of research coming from China, and I personally found it interesting to read, especially as the urban form and demographics are quite different to what is seen in most of the literature in this field which tends to come from the Global North and Australasia. For the most part, the relationships are in the expected directions and the conclusions are appropriate.

It would be good to include a description of the Chinese urban planning policies around provision of access and availability of the infrastructure that is investigated in this paper. Are there any policies that are in place to support equity? What was the distribution of public versus private attributes, and how were these distributed across the two community groupings? Do the low-end and mid-range communities have more/less access to public amenities / facilities? This would be good to unpack this in this paper.

Abstract

Include the n in the abstract, the number of communities investigated, and the year the survey was conducted.

Introduction

Line 43. What do the authors mean by ‘community is a spatial unit of residents’ daily lives’? I’d argue that neighbourhoods are geographically bounded, but communities are socially-produced. For example, there are indigenous communities, LGBTI communities, which do not exist or cannot be operationalised as a spatial unit.

Line 55. Health-effect mechanisms implies a causal relationship, but this study was cross-sectional. Moreover, causal language is used throughout the paper (e.g. line 175, line 447). Please check and remove causal inferences.

Data and methodology

Was the study adequately powered to answer the research question?

How many respondents were from each of the 20 communities? Was the distribution even across the 20 communities?

Line 150: Please justify why these individual attributes were selected.

Line 154: Please justify why these community environment attributes were selected.

Line 160: A weakness of the study is that it uses an 800m Euclidian buffer from centre of community. Furthermore, it was hard to understand how the centre of the community was geographically identified (beyond a round point).

Line 162: What year(s) was the community environment data sourced? Was this temporally matched to the survey?

Line 178 / Table 2: More information is needed about how the quintiles were combined and aggregated to classify the 2 community groupings. Also, would it be more appropriate to have the wording as ‘high-end’, rather than ‘upscale’ community model? That seems to fit better with the comparison group (low-end and mid-range).

Why were low-end and mid-range communities combined? It might be that by combining the groups, important information is missed that could give an indication of built environment associations across the spectrum.

Please state whether ethical approval was granted and by which agency.

Results

Typically, the contextualisation of the results in relation to other research is presented in the discussion section. The results section should only report on the present study’s findings. This paper does not have a discussion section, so it would be helpful to know if the authors have editorial approval to structure the paper in this way. Notwithstanding this comment, I have comments on the results (any amendments would be better placed in the discussion if there is one in the revised manuscript):

Line 342: Could the reason be for the association of poorer health with street intersection density be related to higher air pollution exposure from vehicles stopping and starting?

Line 350: Could the negative relationship between residential density and self-rated health be because residential densities in China are so much greater than other parts of the world, so it brings with it overcrowding, noise pollution, etc?

Table 3. Are the means being presented? If yes, please state this and provide the SD.

Line 400: Suggest dropping the ‘because residents of low-end and mid-range communities are older, have relatively low education, have a higher unemployment rate, and have a relatively lower social status; additionally,’ as it doesn’t add to the point the authors are making.

Discussion

This section currently does not exist in the manuscript.

Study findings need to be contextualised and discussed within the wider body of research in the discussion.

Strengths and weaknesses of the study should be described.

Minor comments

Some of the research cited is quite old.

Something odd is going on with the formatting, whereby some citations have capital letters.

Reviewer 2 Report

Methods
Authors should report the year in which Fifth National Health Services Survey in Wuhan City in which year has been conducted and the year  in which the EQ-5D was measured.
Furthermore, no information are reported about the excluded respondents. How were they distributed between two groups?

Results
Row 358: after constructing the full model the between-group variance of the total sample model decreased by 9.784 and not 9.874 

The Authors mixed the study results with the comparative analysis of the literature.

In order to avoid confusion, the comparative analysis of the literature should be reported separately in a discussion paragraph, in which the study strengths and weakness should be reported also. 

Round 2

Reviewer 1 Report

The authors have done a good job of responding to the queries I raised in my first review. I have a few minor considerations to follow-up on.

  1. Please include in-text the defended use of community (rather than neighbourhood) the authors' included in the rebuttal. I believe the readership will be interested in the municipal structure of Wuhan / China and how communities are conceptualised. It is quite different from a Western perspective.
  2. Please include the institutional ethics approval details in the manuscript - i.e. the approving body, date of approval, and approval number. If the research was not approved by an institute, the research should not have been submitted for publication consideration.
  3. Please include in-text the information provided in the rebuttal regarding justification of the personal and community attributes considered in the research. 
